# Near-linear time approximation algorithms for optimal transport via Sinkhorn iteration

**Jason Altschuler**
MIT
jasonalt@mit.edu

**Jonathan Weed**
MIT
jweed@mit.edu

**Philippe Rigollet**
MIT
rigollet@mit.edu

## Abstract

Computing optimal transport distances such as the earth mover's distance is a fundamental problem in machine learning, statistics, and computer vision. Despite the recent introduction of several algorithms with good empirical performance, it is unknown whether general optimal transport distances can be approximated in near-linear time. This paper demonstrates that this ambitious goal is in fact achieved by Cuturi's *Sinkhorn Distances*. This result relies on a new analysis of Sinkhorn iterations, which also directly suggests a new greedy coordinate descent algorithm GREENKHORN with the same theoretical guarantes. Numerical simulations illustrate that GREENKHORN significantly outperforms the classical SINKHORN algorithm in practice.

*Dedicated to the memory of Michael B. Cohen*

## 1 Introduction

Computing distances between probability measures on metric spaces, or more generally between point clouds, plays an increasingly preponderant role in machine learning [SL11, MJ15, LG15, JSCG16, ACB17], statistics [FCCR16, PZ16, SR04, BGKL17] and computer vision [RTG00, BvdPPH11, SdGP+15]. A prominent example of such distances is the *earth mover's distance* introduced in [WPR85] (see also [RTG00]), which is a special case of Wasserstein distance, or optimal transport (OT) distance [Vil09].

While OT distances exhibit a unique ability to capture geometric features of the objects at hand, they suffer from a heavy computational cost that had been prohibitive in large scale applications until the recent introduction to the machine learning community of *Sinkhorn Distances* by Cuturi [Cut13]. Combined with other numerical tricks, these recent advances have enabled the treatment of large point clouds in computer graphics such as triangle meshes [SdGP+15] and high-resolution neuroimaging data [GPC15]. Sinkhorn Distances rely on the idea of *entropic penalization*, which has been implemented in similar problems at least since Schrödinger [Sch31, Leo14]. This powerful idea has been successfully applied to a variety of contexts not only as a statistical tool for model selection [JRT08, RT11, RT12] and online learning [CBL06], but also as an optimization gadget in first-order optimization methods such as mirror descent and proximal methods [Bub15].

**Related work.** Computing an OT distance amounts to solving the following linear system:

$$\min_{P \in \mathcal{U}_{r,c}} \langle P, C \rangle, \qquad \mathcal{U}_{r,c} := \left\{ P \in \mathbb{R}_+^{n \times n} \, : \, P\mathbf{1} = r, P^\top \mathbf{1} = c \right\}, \qquad (1)$$

where $\mathbf{1}$ is the all-ones vector in $\mathbb{R}^n$, $C \in \mathbb{R}_+^{n \times n}$ is a given *cost matrix*, and $r \in \mathbb{R}^n, c \in \mathbb{R}^n$ are given vectors with positive entries that sum to one. Typically $C$ is a matrix containing pairwise

distances (and is thus dense), but in this paper we allow $C$ to be an arbitrary non-negative dense matrix with bounded entries since our results are more general. For brevity, this paper focuses on square matrices $C$ and $P$, since extensions to the rectangular case are straightforward.

This paper is at the intersection of two lines of research: a theoretical one that aims at finding (near) linear time approximation algorithms for simple problems that are already known to run in polynomial time and a practical one that pursues fast algorithms for solving optimal transport approximately for large datasets.

Noticing that (1) is a linear program with $O(n)$ linear constraints and certain graphical structure, one can use the recent Lee-Sidford linear solver to find a solution in time $\widetilde{O}(n^{2.5})$ [LS14], improving over the previous standard of $O(n^{3.5})$ [Ren88]. While no practical implementation of the Lee-Sidford algorithm is known, it provides a theoretical benchmark for our methods. Their result is part of a long line of work initiated by the seminal paper of Spielman and Teng [ST04] on solving linear systems of equations, which has provided a building block for near-linear time approximation algorithms in a variety of combinatorially structured linear problems. A separate line of work has focused on obtaining faster algorithms for (1) by imposing additional assumptions. For instance, [AS14] obtain approximations to (1) when the cost matrix $C$ arises from a metric, but their running times are not truly near-linear. [SA12, ANOY14] develop even faster algorithms for (1), but require $C$ to arise from a low-dimensional $\ell_p$ metric.

Practical algorithms for computing OT distances include Orlin's algorithm for the *Uncapacitated Minimum Cost Flow* problem via a standard reduction. Like interior point methods, it has a provable complexity of $O(n^3 \log n)$. This dependence on the dimension is also observed in practice, thereby preventing large-scale applications. To overcome the limitations of such general solvers, various ideas ranging from graph sparsification [PW09] to metric embedding [IT03, GD04, SJ08] have been proposed over the years to deal with particular cases of OT distance.

Our work complements both lines of work, theoretical and practical, by providing the first near-linear time guarantee to approximate (1) for general non-negative cost matrices. Moreover we show that this performance is achieved by algorithms that are also very efficient in practice. Central to our contribution are recent developments of scalable methods for general OT that leverage the idea of entropic regularization [Cut13, BCC$^+$15, GCPB16]. However, the apparent practical efficacy of these approaches came without theoretical guarantees. In particular, showing that this regularization yields an algorithm to compute or approximate general OT distances in time nearly linear in the input size $n^2$ was an open question before this work.

**Our contribution.** The contribution of this paper is twofold. First we demonstrate that, with an appropriate choice of parameters, the algorithm for Sinkhorn Distances introduced in [Cut13] is in fact a *near-linear time* approximation algorithm for computing OT distances between discrete measures. This is the first proof that such near-linear time results are achievable for optimal transport. We also provide previously unavailable guidance for parameter tuning in this algorithm. Core to our work is a new and arguably more natural analysis of the Sinkhorn iteration algorithm, which we show converges in a number of iterations independent of the dimension $n$ of the matrix to balance. In particular, this analysis directly suggests a greedy variant of Sinkhorn iteration that also provably runs in near-linear time and significantly outperforms the classical algorithm in practice. Finally, while most approximation algorithms output an approximation of the optimum *value* of the linear program (1), we also describe a simple, parallelizable rounding algorithm that provably outputs a feasible solution to (1). Specifically, for any $\varepsilon > 0$ and bounded, non-negative cost matrix $C$, we describe an algorithm that runs in time $\widetilde{O}(n^2/\varepsilon^3)$ and outputs $\hat{P} \in \mathcal{U}_{r,c}$ such that

$$\langle \hat{P}, C \rangle \leq \min_{P \in \mathcal{U}_{r,c}} \langle P, C \rangle + \varepsilon$$

We emphasize that our analysis does not require the cost matrix $C$ to come from an underlying metric; we only require $C$ to be non-negative. This implies that our results also give, for example, near-linear time approximation algorithms for Wasserstein $p$-distances between discrete measures.

**Notation.** We denote non-negative real numbers by $\mathbb{R}_+$, the set of integers $\{1, \dots, n\}$ by $[n]$, and the $n$-dimensional simplex by $\Delta_n := \{x \in \mathbb{R}_+^n : \sum_{i=1}^n x_i = 1\}$. For two probability distributions $p, q \in \Delta_n$ such that $p$ is absolutely continuous w.r.t. $q$, we define the entropy $H(p)$ of $p$ and the

Kullback-Leibler divergence $\mathcal{K}(p\|q)$ between $p$ and $q$ respectively by

$$H(p) = \sum_{i=1}^{n} p_i \log \left( \frac{1}{p_i} \right), \qquad \mathcal{K}(p\|q) := \sum_{i=1}^{n} p_i \log \left( \frac{p_i}{q_i} \right).$$

Similarly, for a matrix $P \in \mathbb{R}_+^{n \times n}$, we define the entropy $H(P)$ entrywise as $\sum_{ij} P_{ij} \log \frac{1}{P_{ij}}$. We use $\mathbf{1}$ and $\mathbf{0}$ to denote the all-ones and all-zeroes vectors in $\mathbb{R}^n$. For a matrix $A = (A_{ij})$, we denote by $\exp(A)$ the matrix with entries $(e^{A_{ij}})$. For $A \in \mathbb{R}^{n \times n}$, we denote its row and columns sums by $r(A) := A\mathbf{1} \in \mathbb{R}^n$ and $c(A) := A^\top \mathbf{1} \in \mathbb{R}^n$, respectively. The coordinates $r_i(A)$ and $c_j(A)$ denote the $i$th row sum and $j$th column sum of $A$, respectively. We write $\|A\|_\infty = \max_{ij} |A_{ij}|$ and $\|A\|_1 = \sum_{ij} |A_{ij}|$. For two matrices of the same dimension, we denote the Frobenius inner product of $A$ and $B$ by $\langle A, B \rangle = \sum_{ij} A_{ij} B_{ij}$. For a vector $x \in \mathbb{R}^n$, we write $\mathbf{D}(x) \in \mathbb{R}^{n \times n}$ to denote the diagonal matrix with entries $(\mathbf{D}(x))_{ii} = x_i$. For any two nonnegative sequences $(u_n)_n, (v_n)_n$, we write $u_n = \widetilde{O}(v_n)$ if there exist positive constants $C, c$ such that $u_n \leq C v_n (\log n)^c$. For any two real numbers, we write $a \wedge b = \min(a, b)$.

## 2 Optimal Transport in near-linear time

In this section, we describe the main algorithm studied in this paper. Pseudocode appears in Algorithm 1.

The core of our algorithm is the computation of an *approximate Sinkhorn projection* of the matrix $A = \exp(-\eta C)$ (Step 1), details for which will be given in Section 3. Since our approximate Sinkhorn projection is not guaranteed to lie in the feasible set, we round our approximation to ensure that it lies in $\mathcal{U}_{r,c}$ (Step 2). Pseudocode for a simple, parallelizable rounding procedure is given in Algorithm 2.

Algorithm 1 hinges on two subroutines: PROJ and ROUND. We give two algorithms for PROJ: SINKHORN and GREENKHORN. We devote Section 3 to their analysis, which is of independent interest. On the other hand, ROUND is fairly simple. Its analysis is postponed to Section 4.

Our main theorem about Algorithm 1 is the following accuracy and runtime guarantee. The proof is postponed to Section 4, since it relies on the analysis of PROJ and ROUND.

---
**Algorithm 1** APPROXOT$(C, r, c, \varepsilon)$
---
$\eta \leftarrow \frac{4 \log n}{\varepsilon}, \varepsilon' \leftarrow \frac{\varepsilon}{8\|C\|_\infty}$
\\ Step 1: Approximately project onto $\mathcal{U}_{r,c}$
1: $A \leftarrow \exp(-\eta C)$
2: $B \leftarrow \text{PROJ}(A, \mathcal{U}_{r,c}, \varepsilon')$

\\ Step 2: Round to feasible point in $\mathcal{U}_{r,c}$
3: Output $\hat{P} \leftarrow \text{ROUND}(B, \mathcal{U}_{r,c})$

---
**Algorithm 2** ROUND$(F, \mathcal{U}_{r,c})$
---
1: $X \leftarrow \mathbf{D}(x)$ with $x_i = \frac{r_i}{r_i(F)} \wedge 1$
2: $F' \leftarrow XF$
3: $Y \leftarrow \mathbf{D}(y)$ with $y_j = \frac{c_j}{c_j(F')} \wedge 1$
4: $F'' \leftarrow F'Y$
5: $\text{err}_r \leftarrow r - r(F''), \text{err}_c \leftarrow c - c(F'')$
6: Output $G \leftarrow F'' + \text{err}_r \text{err}_c^\top / \|\text{err}_r\|_1$

**Theorem 1.** *Algorithm 1 returns a point $\hat{P} \in \mathcal{U}_{r,c}$ satisfying*

$$\langle \hat{P}, C \rangle \leq \min_{P \in \mathcal{U}_{r,c}} \langle P, C \rangle + \varepsilon$$

*in time $O(n^2 + S)$, where $S$ is the running time of the subroutine $\text{PROJ}(A, \mathcal{U}_{r,c}, \varepsilon')$. In particular, if $\|C\|_\infty \leq L$, then $S$ can be $O(n^2 L^3 (\log n) \varepsilon^{-3})$, so that Algorithm 1 runs in $O(n^2 L^3 (\log n) \varepsilon^{-3})$ time.*

**Remark 1.** *The time complexity in the above theorem reflects only elementary arithmetic operations. In the interest of clarity, we ignore questions of bit complexity that may arise from taking exponentials. The effect of this simplification is marginal since it can be easily shown [KLRS08] that the maximum bit complexity throughout the iterations of our algorithm is $O(L(\log n)/\varepsilon)$. As a result, factoring in bit complexity leads to a runtime of $O(n^2 L^4 (\log n)^2 \varepsilon^{-4})$, which is still truly near-linear.*

# 3 Linear-time approximate Sinkhorn projection

The core of our OT algorithm is the entropic penalty proposed by Cuturi [Cut13]:

$$P_\eta := \underset{P \in \mathcal{U}_{r,c}}{\operatorname{argmin}} \left\{ \langle P, C \rangle - \eta^{-1} H(P) \right\}. \tag{2}$$

The solution to (2) can be characterized explicitly by analyzing its first-order conditions for optimality.

**Lemma 1.** *[Cut13] For any cost matrix $C$ and $r, c \in \Delta_n$, the minimization program (2) has a unique minimum at $P_\eta \in \mathcal{U}_{r,c}$ of the form $P_\eta = XAY$, where $A = \exp(-\eta C)$ and $X, Y \in \mathbb{R}_+^{n \times n}$ are both diagonal matrices. The matrices $(X, Y)$ are unique up to a constant factor.*

We call the matrix $P_\eta$ appearing in Lemma 1 the *Sinkhorn projection* of $A$, denoted $\Pi_{\mathcal{S}}(A, \mathcal{U}_{r,c})$, after Sinkhorn, who proved uniqueness in [Sin67]. Computing $\Pi_{\mathcal{S}}(A, \mathcal{U}_{r,c})$ exactly is impractical, so we implement instead an approximate version $\text{PROJ}(A, \mathcal{U}_{r,c}, \varepsilon')$, which outputs a matrix $B = XAY$ that may not lie in $\mathcal{U}_{r,c}$ but satisfies the condition $\|r(B) - r\|_1 + \|c(B) - c\|_1 \leq \varepsilon'$. We stress that this condition is very natural from a statistical standpoint, since it requires that $r(B)$ and $c(B)$ are close to the target marginals $r$ and $c$ in *total variation distance*.

## 3.1 The classical Sinkhorn algorithm

Given a matrix $A$, Sinkhorn proposed a simple iterative algorithm to approximate the Sinkhorn projection $\Pi_{\mathcal{S}}(A, \mathcal{U}_{r,c})$, which is now known as the Sinkhorn-Knopp algorithm or RAS method. Despite the simplicity of this algorithm and its good performance in practice, it has been difficult to analyze. As a result, recent work showing that $\Pi_{\mathcal{S}}(A, \mathcal{U}_{r,c})$ can be approximated in near-linear time [AZLOW17, CMTV17] has bypassed the Sinkhorn-Knopp algorithm entirely[1]. In our work, we obtain a new analysis of the simple and practical Sinkhorn-Knopp algorithm, showing that it also approximates $\Pi_{\mathcal{S}}(A, \mathcal{U}_{r,c})$ in near-linear time.

Pseudocode for the Sinkhorn-Knopp algorithm appears in Algorithm 3. In brief, it is an alternating projection procedure which renormalizes the rows and columns of $A$ in turn so that they match the desired row and column marginals $r$ and $c$. At each step, it prescribes to either modify all the rows by multiplying row $i$ by $r_i / r_i(A)$ for $i \in [n]$, or to do the analogous operation on the columns. (We interpret the quantity $0/0$ as 1 in this algorithm if ever it occurs.) The algorithm terminates when the matrix $A^{(k)}$ is sufficiently close to the polytope $\mathcal{U}_{r,c}$.

---

**Algorithm 3** $\text{SINKHORN}(A, \mathcal{U}_{r,c}, \varepsilon')$

---
1: Initialize $k \leftarrow 0$
2: $A^{(0)} \leftarrow A/\|A\|_1$, $x^0 \leftarrow \mathbf{0}$, $y^0 \leftarrow \mathbf{0}$
3: **while** $\text{dist}(A^{(k)}, \mathcal{U}_{r,c}) > \varepsilon'$ **do**
4: $\quad k \leftarrow k + 1$
5: $\quad$ **if** $k$ odd **then**
6: $\qquad x_i \leftarrow \log \frac{r_i}{r_i(A^{(k-1)})}$ for $i \in [n]$
7: $\qquad x^k \leftarrow x^{k-1} + x, \, y^k \leftarrow y^{k-1}$
8: $\quad$ **else**
9: $\qquad y \leftarrow \log \frac{c_j}{c_j(A^{(k-1)})}$ for $j \in [n]$
10: $\qquad y^k \leftarrow y^{k-1} + y, \, x^k \leftarrow x^{k-1}$
11: $\quad A^{(k)} = \mathbf{D}(\exp(x^k)) A \mathbf{D}(\exp(y^k))$
12: Output $B \leftarrow A^{(k)}$

---

## 3.2 Prior work

Before this work, the best analysis of Algorithm 3 showed that $\widetilde{O}((\varepsilon')^{-2})$ iterations suffice to obtain a matrix close to $\mathcal{U}_{r,c}$ in $\ell_2$ distance:

**Proposition 1.** *[KLRS08] Let $A$ be a strictly positive matrix. Algorithm 3 with $\text{dist}(A, \mathcal{U}_{r,c}) = \|r(A) - r\|_2 + \|c(A) - c\|_2$ outputs a matrix $B$ satisfying $\|r(B) - r\|_2 + \|c(B) - c\|_2 \leq \varepsilon'$ in $O(\rho(\varepsilon')^{-2} \log(s/\ell))$ iterations, where $s = \sum_{ij} A_{ij}$, $\ell = \min_{ij} A_{ij}$, and $\rho > 0$ is such that $r_i, c_i \leq \rho$ for all $i \in [n]$.*

Unfortunately, this analysis is not strong enough to obtain a true near-linear time guarantee. Indeed, the $\ell_2$ norm is not an appropriate measure of closeness between probability vectors, since very different distributions on large alphabets can nevertheless have small $\ell_2$ distance: for example, $(n^{-1}, \ldots, n^{-1}, 0, \ldots, 0)$ and $(0, \ldots, 0, n^{-1}, \ldots, n^{-1})$ in $\Delta_{2n}$ have $\ell_2$ distance $\sqrt{2/n}$ even though

they have disjoint support. As noted above, for statistical problems, including computation of the OT distance, it is more natural to measure distance in $\ell_1$ norm.

The following Corollary gives the best $\ell_1$ guarantee available from Proposition 1.

**Corollary 1.** *Algorithm 3 with* $\mathrm{dist}(A,\mathcal{U}_{r,c}) = \|r(A) - r\|_2 + \|c(A) - c\|_2$ *outputs a matrix* $B$ *satisfying* $\|r(B) - r\|_1 + \|c(B) - c\|_1 \leq \varepsilon'$ *in* $O\big(n\rho(\varepsilon')^{-2}\log(s/\ell)\big)$ *iterations.*

The extra factor of $n$ in the runtime of Corollary 1 is the price to pay to convert an $\ell_2$ bound to an $\ell_1$ bound. Note that $\rho \geq 1/n$, so $n\rho$ is always larger than 1. If $r = c = \mathbf{1}_n/n$ are uniform distributions, then $n\rho = 1$ and no dependence on the dimension appears. However, in the extreme where $r$ or $c$ contains an entry of constant size, we get $n\rho = \Omega(n)$.

### 3.3 New analysis of the Sinkhorn algorithm

Our new analysis allows us to obtain a dimension-independent bound on the number of iterations beyond the uniform case.

**Theorem 2.** *Algorithm 3 with* $\mathrm{dist}(A,\mathcal{U}_{r,c}) = \|r(A) - r\|_1 + \|c(A) - c\|_1$ *outputs a matrix* $B$ *satisfying* $\|r(B) - r\|_1 + \|c(B) - c\|_1 \leq \varepsilon'$ *in* $O\big((\varepsilon')^{-2}\log(s/\ell)\big)$ *iterations, where* $s = \sum_{ij} A_{ij}$ *and* $\ell = \min_{ij} A_{ij}$.

Comparing our result with Corollary 1, we see what our bound is always stronger, by up to a factor of $n$. Moreover, our analysis is extremely short. Our improved results and simplified proof follow directly from the fact that we carry out the analysis entirely with respect to the Kullback–Leibler divergence, a common measure of statistical distance. This measure possesses a close connection to the total-variation distance via Pinsker's inequality (Lemma 4, below), from which we obtain the desired $\ell_1$ bound. Similar ideas can be traced back at least to [GY98] where an analysis of Sinkhorn iterations for bistochastic targets is sketched in the context of a different problem: detecting the existence of a perfect matching in a bipartite graph.

We first define some notation. Given a matrix $A$ and desired row and column sums $r$ and $c$, we define the potential (Lyapunov) function $f : \mathbb{R}^n \times \mathbb{R}^n \to \mathbb{R}$ by

$$f(x,y) = \sum_{ij} A_{ij} e^{x_i + y_j} - \langle r, x \rangle - \langle c, y \rangle \,.$$

This auxiliary function has appeared in much of the literature on Sinkhorn projections [KLRS08, CMTV17, KK96, KK93]. We call the vectors $x$ and $y$ *scaling vectors*. It is easy to check that a minimizer $(x^*, y^*)$ of $f$ yields the Sinkhorn projection of $A$: writing $X = \mathbf{D}(\exp(x^*))$ and $Y = \mathbf{D}(\exp(y^*))$, first order optimality conditions imply that $XAY$ lies in $\mathcal{U}_{r,c}$, and therefore $XAY = \Pi_{\mathcal{S}}(A,\mathcal{U}_{r,c})$.

The following lemma exactly characterizes the improvement in the potential function $f$ from an iteration of Sinkhorn, in terms of our current divergence to the target marginals.

**Lemma 2.** *If* $k \geq 2$, *then* $f(x^{k-1}, y^{k-1}) - f(x^k, y^k) = \mathcal{K}(r\|r(A^{(k-1)})) + \mathcal{K}(c\|c(A^{(k-1)}))$.

*Proof.* Assume without loss of generality that $k$ is odd, so that $c(A^{(k-1)}) = c$ and $r(A^{(k)}) = r$. (If $k$ is even, interchange the roles of $r$ and $c$.) By definition,

$$f(x^{k-1}, y^{k-1}) - f(x^k, y^k) = \sum_{ij} \big(A_{ij}^{(k-1)} - A_{ij}^{(k)}\big) + \langle r, x^k - x^{k-1} \rangle + \langle c, y^k - y^{k-1} \rangle$$

$$= \sum_i r_i(x_i^k - x_i^{k-1}) = \mathcal{K}(r\|r(A^{(k-1)})) + \mathcal{K}(c\|c(A^{(k-1)})) \,,$$

where we have used that: $\|A^{(k-1)}\|_1 = \|A^{(k)}\|_1 = 1$ and $Y^{(k)} = Y^{(k-1)}$; for all $i$, $r_i(x_i^k - x_i^{k-1}) = r_i \log \frac{r_i}{r_i(A^{(k-1)})}$; and $\mathcal{K}(c\|c(A^{(k-1)})) = 0$ since $c = c(A^{(k-1)})$. $\qquad \square$

The next lemma has already appeared in the literature and we defer its proof to the supplement.

**Lemma 3.** *If* $A$ *is a positive matrix with* $\|A\|_1 \leq s$ *and smallest entry* $\ell$, *then*

$$f(x^1, y^1) - \min_{x,y \in \mathbb{R}} f(x,y) \leq f(0,0) - \min_{x,y \in \mathbb{R}} f(x,y) \leq \log \frac{s}{\ell} \,.$$

**Lemma 4** (Pinsker's Inequality). *For any probability measures $p$ and $q$, $\|p - q\|_1 \leq \sqrt{2\mathcal{K}(p\|q)}$.*

*Proof of Theorem 2.* Let $k^*$ be the first iteration such that $\|r(A^{(k^*)}) - r\|_1 + \|c(A^{(k^*)}) - c\|_1 \leq \varepsilon'$. Pinsker's inequality implies that for any $k < k^*$, we have

$$\varepsilon'^2 < (\|r(A^{(k)}) - r\|_1 + \|c(A^{(k)}) - c\|_1)^2 \leq 4(\mathcal{K}(r\|r(A^{(k)}) + \mathcal{K}(c\|c(A^{(k)}))),$$

so Lemmas 2 and 3 imply that we terminate in $k^* \leq 4\varepsilon'^{-2} \log(s/\ell)$ steps, as claimed. $\qquad\square$

### 3.4 Greedy Sinkhorn

In addition to a new analysis of SINKHORN, we propose a new algorithm GREENKHORN which enjoys the same convergence guarantee but performs better in practice. Instead of performing alternating updates of *all* rows and columns of $A$, the GREENKHORN algorithm updates only a *single* row or column at each step. Thus GREENKHORN updates only $O(n)$ entries of $A$ per iteration, rather than $O(n^2)$.

In this respect, GREENKHORN is similar to the stochastic algorithm for Sinkhorn projection proposed by [GCPB16]. There is a natural interpretation of both algorithms as coordinate descent algorithms in the dual space corresponding to row/column violations. Nevertheless, our algorithm differs from theirs in several key ways. Instead of choosing a row or column to update randomly, GREENKHORN chooses the best row or column to update greedily. Additionally, GREENKHORN does an exact line search on the coordinate in question since there is a simple closed form for the optimum, whereas the algorithm proposed by [GCPB16] updates in the direction of the average gradient. Our experiments establish that GREENKHORN performs better in practice; more details appear in the Supplement.

We emphasize that although this algorithm is an extremely natural modification of SINKHORN, previous analyses of SINKHORN cannot be modified to extract any meaningful performance guarantees on GREENKHORN. On the other hand, our new analysis of SINKHORN from Section 3.3 applies to GREENKHORN with only trivial modifications.

---

**Algorithm 4** GREENKHORN$(A, \mathcal{U}_{r,c}, \varepsilon')$

1: $A^{(0)} \leftarrow A/\|A\|_1$, $x \leftarrow \mathbf{0}$, $y \leftarrow \mathbf{0}$.
2: $A \leftarrow A^{(0)}$
3: **while** $\text{dist}(A, \mathcal{U}_{r,c}) > \varepsilon$ **do**
4: $\quad I \leftarrow \text{argmax}_i \, \rho(r_i, r_i(A))$
5: $\quad J \leftarrow \text{argmax}_j \, \rho(c_j, c_j(A))$
6: $\quad$ **if** $\rho(r_I, r_I(A)) > \rho(c_J, c_J(A))$ **then**
7: $\quad\quad x_I \leftarrow x_I + \log \frac{r_I}{r_I(A)}$
8: $\quad$ **else**
9: $\quad\quad y_J \leftarrow y_J + \log \frac{c_J}{c_J(A)}$
10: $\quad A \leftarrow \mathbf{D}(\exp(x))A^{(0)}\mathbf{D}(\exp(y))$
11: Output $B \leftarrow A$

---

Pseudocode for GREENKHORN appears in Algorithm 4. We let $\text{dist}(A, \mathcal{U}_{r,c}) = \|r(A) - r\|_1 + \|c(A) - c\|_1$ and define the distance function $\rho : \mathbb{R}_+ \times \mathbb{R}_+ \to [0, +\infty]$ by

$$\rho(a, b) = b - a + a \log \frac{a}{b}.$$

The choice of $\rho$ is justified by its appearance in Lemma 5, below. While $\rho$ is not a metric, it is easy to see that $\rho$ is nonnegative and satisfies $\rho(a, b) = 0$ iff $a = b$.

We note that after $r(A)$ and $c(A)$ are computed once at the beginning of the algorithm, GREENKHORN can easily be implemented such that each iteration runs in only $O(n)$ time.

**Theorem 3.** *The algorithm GREENKHORN outputs a matrix $B$ satisfying $\|r(B) - r\|_1 + \|c(B) - c\|_1 \leq \varepsilon'$ in $O(n(\varepsilon')^{-2} \log(s/\ell))$ iterations, where $s = \sum_{ij} A_{ij}$ and $\ell = \min_{ij} A_{ij}$. Since each iteration takes $O(n)$ time, such a matrix can be found in $O(n^2(\varepsilon')^{-2} \log(s/\ell))$ time.*

The analysis requires the following lemma, which is an easy modification of Lemma 2.

**Lemma 5.** *Let $A'$ and $A''$ be successive iterates of GREENKHORN, with corresponding scaling vectors $(x', y')$ and $(x'', y'')$. If $A''$ was obtained from $A'$ by updating row $I$, then*

$$f(x', y') - f(x'', y'') = \rho(r_I, r_I(A')),$$

*and if it was obtained by updating column $J$, then*

$$f(x', y') - f(x'', y'') = \rho(c_J, c_J(A')).$$

We also require the following extension of Pinsker's inequality (proof in Supplement).

**Lemma 6.** *For any $\alpha \in \Delta_n, \beta \in \mathbb{R}_+^n$, define $\rho(\alpha, \beta) = \sum_i \rho(\alpha_i, \beta_i)$. If $\rho(\alpha, \beta) \leq 1$, then*

$$\|\alpha - \beta\|_1 \leq \sqrt{7\rho(\alpha, \beta)} \, .$$

*Proof of Theorem 3.* We follow the proof of Theorem 2. Since the row or column update is chosen greedily, at each step we make progress of at least $\frac{1}{2n}(\rho(r, r(A)) + \rho(c, c(A)))$. If $\rho(r, r(A))$ and $\rho(c, c(A))$ are both at most 1, then under the assumption that $\|r(A) - r\|_1 + \|c(A) - c\|_1 > \varepsilon'$, our progress is at least

$$\frac{1}{2n}(\rho(r, r(A)) + \rho(c, c(A))) \geq \frac{1}{14n}(\|r(A) - r\|_1^2 + \|c(A) - c\|_1^2) \geq \frac{1}{28n}\varepsilon'^2$$

Likewise, if either $\rho(r, r(A))$ or $\rho(c, c(A))$ is larger than 1, our progress is at least $1/2n \geq \frac{1}{28n}\varepsilon'^2$. Therefore, we terminate in at most $28n\varepsilon'^{-2} \log(s/\ell)$ iterations. $\square$

# 4 Proof of Theorem 1

First, we present a simple guarantee about the rounding Algorithm 2. The following lemma shows that the $\ell_1$ distance between the input matrix $F$ and rounded matrix $G = \text{ROUND}(F, \mathcal{U}_{r,c})$ is controlled by the total-variation distance between the input matrix's marginals $r(F)$ and $c(F)$ and the desired marginals $r$ and $c$.

**Lemma 7.** *If $r, c \in \Delta_n$ and $F \in \mathbb{R}_+^{n \times n}$, then Algorithm 2 takes $O(n^2)$ time to output a matrix $G \in \mathcal{U}_{r,c}$ satisfying*

$$\|G - F\|_1 \leq 2\Big[\|r(F) - r\|_1 + \|c(F) - c\|_1\Big] \, .$$

The proof of Lemma 7 is simple and left to the Supplement. (We also describe in the Supplement a randomized variant of Algorithm 2 that achieves a slightly better bound than Lemma 7). We are now ready to prove Theorem 1.

*Proof of Theorem 1.* ERROR ANALYSIS. Let $B$ be the output of $\text{PROJ}(A, \mathcal{U}_{r,c}, \varepsilon')$, and let $P^* \in \text{argmin}_{P \in \mathcal{U}_{r,c}}\langle P, C\rangle$ be an optimal solution to the original OT program.

We first show that $\langle B, C\rangle$ is not much larger than $\langle P^*, C\rangle$. To that end, write $r' := r(B)$ and $c' := c(B)$. Since $B = XAY$ for positive diagonal matrices $X$ and $Y$, Lemma 1 implies $B$ is the optimal solution to

$$\min_{P \in \mathcal{U}_{r',c'}} \langle P, C\rangle - \eta^{-1}H(P) \, . \tag{3}$$

By Lemma 7, there exists a matrix $P' \in \mathcal{U}_{r',c'}$ such that $\|P' - P^*\|_1 \leq 2(\|r' - r\|_1 + \|c' - c\|_1)$. Moreover, since $B$ is an optimal solution of (3), we have

$$\langle B, C\rangle - \eta^{-1}H(B) \leq \langle P', C\rangle - \eta^{-1}H(P') \, .$$

Thus, by Hölder's inequality

$$\begin{aligned}
\langle B, C\rangle - \langle P^*, C\rangle &= \langle B, C\rangle - \langle P', C\rangle + \langle P', C\rangle - \langle P^*, C\rangle \\
&\leq \eta^{-1}(H(B) - H(P')) + 2(\|r' - r\|_1 + \|c' - c\|_1)\|C\|_\infty \\
&\leq 2\eta^{-1}\log n + 2(\|r' - r\|_1 + \|c' - c\|_1)\|C\|_\infty \, ,
\end{aligned} \tag{4}$$

where we have used the fact that $0 \leq H(B), H(P') \leq 2\log n$.

Lemma 7 implies that the output $\hat{P}$ of $\text{ROUND}(B, \mathcal{U}_{r,c})$ satisfies the inequality $\|B - \hat{P}\|_1 \leq 2(\|r' - r\|_1 + \|c' - c\|_1)$. This fact together with (4) and Hölder's inequality yields

$$\langle \hat{P}, C\rangle \leq \min_{P \in \mathcal{U}_{r,c}} \langle P, C\rangle + 2\eta^{-1}\log n + 4(\|r' - r\|_1 + \|c' - c\|_1)\|C\|_\infty \, .$$

Applying the guarantee of $\text{PROJ}(A, \mathcal{U}_{r,c}, \varepsilon')$, we obtain

$$\langle \hat{P}, C\rangle \leq \min_{P \in \mathcal{U}_{r,c}} \langle P, C\rangle + \frac{2\log n}{\eta} + 4\varepsilon'\|C\|_\infty \, .$$

Plugging in the values of $\eta$ and $\varepsilon'$ prescribed in Algorithm 1 finishes the error analysis.

RUNTIME ANALYSIS. Lemma 7 shows that Step 2 of Algorithm 1 takes $O(n^2)$ time. The runtime of Step 1 is dominated by the PROJ$(A, \mathcal{U}_{r,c}, \varepsilon')$ subroutine. Theorems 2 and 3 imply that both the SINKHORN and GREENKHORN algorithms accomplish this in $S = O(n^2(\varepsilon')^{-2} \log \frac{s}{\ell})$ time, where $s$ is the sum of the entries of $A$ and $\ell$ is the smallest entry of $A$. Since the matrix $C$ is nonnegative, the entries of $A$ are bounded above by 1, thus $s \leq n^2$. The smallest entry of $A$ is $e^{-\eta\|C\|_\infty}$, so $\log 1/\ell = \eta\|C\|_\infty$. We obtain $S = O(n^2(\varepsilon')^{-2}(\log n + \eta\|C\|_\infty))$. The proof is finished by plugging in the values of $\eta$ and $\varepsilon'$ prescribed in Algorithm 1. $\qquad\square$

## 5 Empirical results

Cuturi [Cut13] already gave experimental evidence that using SINKHORN to solve (2) outperforms state-of-the-art techniques for optimal transport. In this section, we provide strong empirical evidence that our proposed GREENKHORN algorithm significantly outperforms SINKHORN.

We consider transportation between pairs of $m \times m$ greyscale images, normalized to have unit total mass. The target marginals $r$ and $c$ represent two images in a pair, and $C \in \mathbb{R}^{m^2 \times m^2}$ is the matrix of $\ell_1$ distances between pixel locations. Therefore, we aim to compute the earth mover's distance.

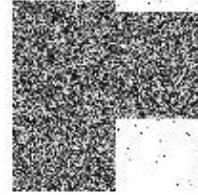

Figure 1: Synthetic image.

We run experiments on two datasets: *real images*, from MNIST, and *synthetic images*, as in Figure 1.

### 5.1 MNIST

We first compare the behavior of GREENKHORN and SINKHORN on real images. To that end, we choose 10 random pairs of images from the MNIST dataset, and for each one analyze the performance of APPROXOT when using both GREENKHORN and SINKHORN for the approximate projection step. We add negligible noise $0.01$ to each background pixel with intensity $0$. Figure 2 paints a clear picture: GREENKHORN significantly outperforms SINKHORN both in the short and long term.

### 5.2 Random images

To better understand the empirical behavior of both algorithms in a number of different regimes, we devised a synthetic and tunable framework whereby we generate images by choosing a randomly positioned "foreground" square in an otherwise black background. The size of this square is a tunable parameter varied between 20%, 50%, and 80% of the total image's area. Intensities of background pixels are drawn uniformly from $[0, 1]$; foreground pixels are drawn uniformly from $[0, 50]$. Such an image is depicted in Figure 1, and results appear in Figure 2.

We perform two other experiments with random images in Figure 3. In the first, we vary the number of background pixels and show that GREENKHORN performs better when the number of background pixels is

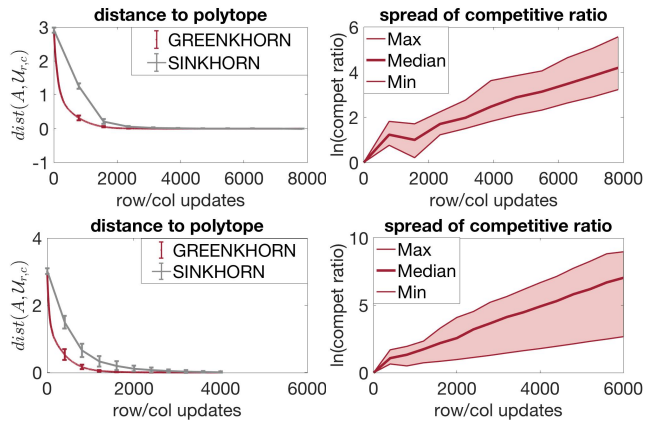

Figure 2: Comparison of GREENKHORN and SINKHORN on pairs of MNIST images of dimension $28 \times 28$ (top) and random images of dimension $20 \times 20$ with 20% foreground (bottom). Left: distance $\mathrm{dist}(A, \mathcal{U}_{r,c})$ to the transport polytope (average over 10 random pairs of images). Right: maximum, median, and minimum values of the competitive ratio $\ln(\mathrm{dist}(A_S, \mathcal{U}_{r,c})/\mathrm{dist}(A_G, \mathcal{U}_{r,c}))$ over 10 runs.

larger. We conjecture that this is related to the fact that GREENKHORN only updates salient rows and

columns at each step, whereas SINKHORN wastes time updating rows and columns corresponding to background pixels, which have negligible impact. This demonstrates that GREENKHORN is a better choice especially when data is sparse, which is often the case in practice.

In the second, we consider the role of the regularization parameter $\eta$. Our analysis requires taking $\eta$ of order $\log n/\varepsilon$, but Cuturi [Cut13] observed that in practice $\eta$ can be much smaller. Cuturi showed that SINKHORN outperforms state-of-the art techniques for computing OT distance even when $\eta$ is a small constant, and Figure 3 shows that GREENKHORN runs faster than SINKHORN in this regime with no loss in accuracy.

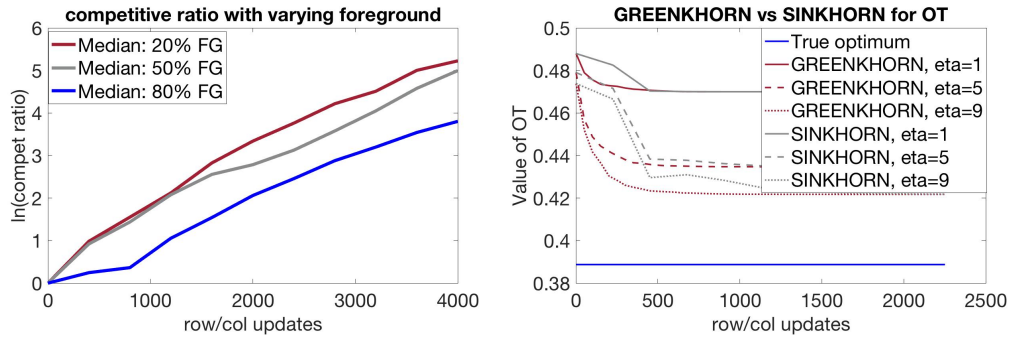

Figure 3: Left: Comparison of median competitive ratio for random images containing 20%, 50%, and 80% foreground. Right: Performance of GREENKHORN and SINKHORN for small values of $\eta$.

## Acknowledgments

We thank Michael Cohen, Adrian Vladu, John Kelner, Justin Solomon, and Marco Cuturi for helpful discussions. We are grateful to Pablo Parrilo for drawing our attention to the fact that GREENKHORN is a coordinate descent algorithm, and to Alexandr Andoni for references.

JA and JW were generously supported by NSF Graduate Research Fellowship 1122374. PR is supported in part by grants NSF CAREER DMS-1541099, NSF DMS-1541100, NSF DMS-1712596, DARPA W911NF-16-1-0551, ONR N00014-17-1-2147 and a grant from the MIT NEC Corporation.

## Footnotes

[1]Replacing the PROJ step in Algorithm 1 with the matrix-scaling algorithm developed in [CMTV17] results in a runtime that is a single factor of $\varepsilon$ faster than what we present in Theorem 1. The benefit of our approach is that it is extremely easy to implement, whereas the matrix-scaling algorithm of [CMTV17] relies heavily on near-linear time Laplacian solver subroutines, which are not implementable in practice.

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
