[Supplementary Material]

# Supplement to "Near-linear time approximation algorithms for optimal transport via Sinkhorn iteration"

**Jason Altschuler**
MIT
jasonalt@mit.edu

**Jonathan Weed**
MIT
jweed@mit.edu

**Philippe Rigollet**
MIT
rigollet@mit.edu

## 1 Proof of Lemma 3

The proof of the first inequality is similar to the proof of Lemma 2:

$$f(0,0) - f(x^{(1)}, y^{(1)}) = \langle r, x^{(1)} \rangle + \langle c, y^{(1)} \rangle = \sum_{ij} A_{ij}^{(1)} \log \frac{A_{ij}^{(1)}}{A_{ij}^{(0)}} = \mathcal{K}(A^{(1)} \| A^{(0)}) \geq 0 \,,$$

where $\mathcal{K}(A^{(1)} \| A^{(0)})$ denotes the divergence between $A^{(1)}$ and $A^{(0)}$ viewed as elements of $\Delta_{n^2}$.

We now prove the second claim. Note that $A^{(0)}$ satisfies $\|A^{(0)}\|_1 = 1$ and has smallest entry $\ell/s$. Since $A^{(0)}$ is positive, [Sin67] shows that $\Pi_{\mathcal{S}}(A^{(0)})$ exists and is unique. Let $(x^*, y^*)$ be corresponding scaling factors. Then

$$f(0,0) - f(x^*, y^*) = \langle r, x^* \rangle + \langle c, y^* \rangle \,.$$

Now since

$$A_{ij}^{(0)} e^{x_i^* + y_j^*} \leq \sum_{ij} A_{ij}^{(0)} e^{x_i^* + y_j^*} = 1 \,,$$

we have

$$x_i^* + y_j^* \leq \log \frac{s}{\ell} \,,$$

for all $i, j \in [n]$. Thus because $r$ and $c$ are both probability vectors,

$$\langle r, x^* \rangle + \langle c, y^* \rangle \leq \log \frac{s}{\ell} \,.$$

$\square$

## 2 Proof of Lemma 5

We prove only the case where a row was updated, since the column case is exactly the same.

By definition,

$$f(x', y') - f(x'', y'') = \sum_{ij} (A_{ij}' - A_{ij}'') + \langle r, x'' - x' \rangle + \langle c, y'' - y' \rangle \,.$$

Observe that $A'$ and $A''$ differ only in the $I$th row, and $x''$ and $x'$ differ only in the $I$th entry, and $y'' = y'$. Hence

$$f(x', y') - f(x'', y'') = r_I(A') - r_I(A'') + r_I(x_I'' - x_I')$$
$$= \rho(r_I, r_I(A')) \,,$$

where we have used the fact that $r_I(A'') = r_I$ and $x_I'' - x_I' = \log(r_I/r_I(A'))$.

$\square$

# 3 Proof of Lemma 6

Let $s = \sum_i \beta_i$, and write $\bar{\beta} = \beta/s$. The definition of $\rho$ implies

$$\rho(\alpha, \beta) = \sum_i (\beta_i - \alpha_i) + \alpha_i \log \frac{\alpha_i}{\beta_i}$$

$$= s - 1 + \sum_i \alpha_i \log \frac{\alpha_i}{s\bar{\beta}_i}$$

$$= s - 1 - (\log s) \sum_i \alpha_i + \mathcal{K}(\alpha \| \bar{\beta})$$

$$= s - 1 - \log s + \mathcal{K}(\alpha \| \bar{\beta}).$$

Note that both $s - 1 - \log s$ and $\mathcal{K}(\alpha \| \bar{\beta})$ are nonnegative. If $\rho(\alpha, \beta) \leq 1$, then in particular $s - 1 - \log s \leq 1$, and it can be seen that $s - 1 - \log s \geq (s-1)^2/5$ in this range. Applying Lemma 4 (Pinsker's inequality) yields

$$\rho(\alpha, \beta) \geq \frac{1}{5}(s-1)^2 + \frac{1}{2}\|\alpha - \bar{\beta}\|_1^2.$$

By the triangle inequality and convexity,

$$\|\alpha - \beta\|_1^2 \leq (\|\bar{\beta} - \beta\|_1 + \|\alpha - \bar{\beta}\|_1)^2 = (|s-1| + \|\alpha - \bar{\beta}\|_1)^2 \leq \frac{7}{5}(s-1)^2 + \frac{7}{2}\|\alpha - \bar{\beta}\|_1^2.$$

The claim follows from the above two displays. $\square$

# 4 Proof of Lemma 7

Let $G$ be the output of $\textsc{round}(F, \mathcal{U}_{r,c})$. The entries of $F''$ are nonnegative, and at the end of the algorithm $\text{err}_r$ and $\text{err}_c$ are both nonnegative, with $\|\text{err}_r\|_1 = \|\text{err}_c\|_1 = 1 - \|F''\|_1$. Therefore the entries of $G$ are nonnegative and

$$r(G) = r(F'') + r(\text{err}_r \text{err}_c^\top / \|\text{err}_r\|_1) = r(F'') + \text{err}_r = r,$$

and likewise $c(G) = c$. This establishes that $G \in \mathcal{U}_{r,c}$.

Now we prove the $\ell_1$ bound between the original matrix $F$ and $G$. Let $\Delta = \|F\|_1 - \|F''\|_1$ be the total amount of mass removed from $F$ by rescaling the rows and columns. In the first step, we remove mass from a row of $F$ when $r_i(F) \geq r_i$, and in the second step we remove mass from a column when $c_j(F') \geq c_j$. We therefore have

$$\Delta = \sum_{i=1}^n (r_i(F) - r_i)_+ + \sum_{j=1}^n (c_j(F') - c_j)_+. \tag{1}$$

Let us analyze both of the sums in (1). First, a simple calculation shows

$$\sum_{i=1}^n (r_i(F) - r_i)_+ = \frac{1}{2}\Big[\|r(F) - r\|_1 + \|F\| - 1\Big].$$

Next, upper bound the second sum in (1) using the fact that the vector $c(F)$ is entrywise larger than $c(F')$

$$\sum_{j=1}^n (c_j(F') - c_j)_+ \leq \sum_{j=1}^n (c_j(F) - c_j)_+ \leq \|c(F) - c\|_1$$

Therefore we conclude

$$\|G - F\|_1 \leq \Delta + \|\text{err}_r \text{err}_c^\top\|_1 / \|\text{err}_r\|_1$$

$$= \Delta + 1 - \|F''\|_1$$

$$= 2\Delta + 1 - \|F\|_1$$

$$\leq \|r(F) - r\|_1 + 2\|c(F) - c\|_1 \tag{2}$$

$$\leq 2\Big[\|r(F) - r\|_1 + \|c(F) - c\|_1\Big]$$

Finally, we prove the $O(n^2)$ runtime bound follows by observing that each rescaling and computing the matrix $\text{err}_r \text{err}_c^\top / \|\text{err}_r\|_1$ both require at most $O(n^2)$ time. $\square$

# 5    Randomized variant of rounding algorithm (Algorithm 2)

In the section, we describe a simple randomized variant of Algorithm 2 that achieves a slightly better guarantee. Let us first recall the guarantee we get for Algorithm 2. By equation (2) in the proof of Lemma 7, the $\ell_1$ difference between the original matrix $F$ and rounded matrix $G$ is upper bounded by

$$\|G - F\|_1 \leq \|r(F) - r\|_1 + 2\|c(F) - c\|_1 \,.$$

This asymmetry between $\|r(F) - r\|_1$ and $\|c(F) - c\|_1$ arises because Algorithm 2 creates $F''$ by first removing mass from rows of $F$, and then from columns. Consider modifying Algorithm 2 to create $F''$ by first removing mass from columns of $F$, and then from rows. Then a symmetrical argument gives the bound

$$\|G - F\|_1 \leq 2\|r(F) - r\|_1 + \|c(F) - c\|_1 \,.$$

Together the above two displays suggest the following simple randomized variant of Algorithm 2: with probability $1/2$, perform Algorithm 2; otherwise, perform the above-described column-then-row version of Algorithm 2. Combining the above two displays then gives the following improved bound for this randomized algorithm

$$\mathbb{E}\|G - F\|_1 \leq \frac{3}{2}\Big[\|r(F) - r\|_1 + \|c(F) - c\|_1\Big] \,.$$

# 6    Comparison with [GCPB16]

In this Section, we present an empirical comparison of the performance of GREENKHORN with the stochastic algorithm proposed by [GCPB16]. Their algorithm—which we call Stochastic Sinkhorn for convenience—uses a Stochastic Averaged Gradient (SAG) algorithm to optimize a dual version of the entropic penalty program (2).

We have noted in the main text that GREENKHORN and Stochastic Sinkhorn both attempt to solve the scaling problem via coordinate descent in the dual problem. Stochastic Sinkhorn does so via the method proposed in [SLRB17], whereas GREENKHORN greedily chooses a good coordinate to update, and then leverages an explicit closed form to perform an exact line search on this coordinate. One difference between our algorithms is their starting point: GREENKHORN is initialized with $A/\|A\|_1$, whereas the starting primal solution corresponding to the initialization of Stochastic Sinkhorn is the matrix obtained by first multiplying each column of $A$ by the corresponding entry of $c$ and then scaling the rows of the resulting matrix so they agree with $r$. This is equivalent to performing a full update step of SINKHORN on the matrix $A\mathbf{D}(c)$ at the beginning of this algorithm. In simulations, this starting point is of better quality than the matrix $A/\|A\|_1$ which GREENKHORN uses as its first iterate; however, this advantage quickly disappears. Since our goal is to compare GREENKHORN and Stochastic Sinkhorn in terms of the number of required row or column updates, we also initialize GREENKHORN at this point instead of at $A/\|A\|_1$ to facilitate an apples-to-apples comparison.

To compare the performance of GREENKHORN with Stochastic Sinkhorn, we use an experiment on random images with 20% foreground pixels, as in Section 5.2. We initialize both algorithms with the same primal solution and used Algorithm 2 to round iterates of each algorithm to the feasible polytope $\mathcal{U}_{r,c}$. Implementing Stochastic Sinkhorn requires choosing a step size, denoted by $C$ in [GCPB16]. That papers suggests choosing $C = 1/(Ln)$, $3/(Ln)$, or $5/(Ln)$, where $L$ is an upper bound on the Lipschitz constant of the semi-dual problem they consider.[1] We compare all three choices of step size with our implementation of the GREENKHORN algorithm in Figure 1 with two different values of the parameter $\eta$.

Figure 1: Comparison of GREENKHORN and Stochastic Sinkhorn

## Footnotes

[1]In fact, they propose the step sizes $C = 1/L, 3/L, 5/L$ in the main text, but the extra factor of $n$ is present in the simulation code posted online, so we have opted to retain it in our experiments. Our experimental results indicate that without the factor of $n$, the resulting algorithm is quite unstable.