[Reviews · NeurIPS 2017]

Reviewer 1



This paper proposes an algorithm for solving unregularized OT problems. The two-step algorithm (Sinkhorn + rounding) obtains an epsilon-accurate approximation in O(n^2 log n / epsilon^3) time. The authors further propose a greedy variant of Sinkhorn. I am not an expert in the field but this paper seems like an important contribution. The paper also includes a new analysis of Sinkhorn where the distance to the transportation polytope is measured in total variation, which as the authors argue is more relevant. I found it a bit unfair that the authors talk about cubic dependence in line 48 for interior points methods but when it comes to their method talk about linear dependence in n^2. In fact, I found the near-linear "time" claims in the title and in the paper misleading. IMO, near linearity is w.r.t. the cost matrix size. The "rounding to a feasible point" section was a bit too short. It was not entirely clear to me what calculating "the optimal transport with respect to the total variation distance" means. What precise objective (if any) does Algorithm 2 minimize? The cubic dependence on epsilon and on L suggests that there are regimes where network flow solvers are faster, in addition to being exact. This should be properly discussed. The empirical evidence that Greenkhorn outperforms Sinkhorn seems a bit anecdotal. In their experiments, the authors plot their results against the number of row / col updates. However, on GPU, the cost of updating one or several rows / cols should likely be more or less the same. I would have been more convinced by wall-clock time comparison. More evidence on more datasets, possibly in the supplementary, would be welcome. Minor comments -------------- Line 6: "provides guidance towards parameter tuning for this algorithm" I am not sure whether this is really relevant for the abstract. This is not really discussed in the paper and the user has to choose epsilon anyway. Line 9: mention that Greenkhorn is greedy Line 72: perhaps mention that the solution is only feasible asymptotically? Line 89: progam -> program Line 101 (Theorem 3): epsilon -> epsilon'? Theorem 5: if space permits, repeat the definitions of s and \ell Line 231: give a brief justification / intuition for the choice of rho(a, b)

Reviewer 2



In this paper the authors discuss entropic regularized optimal transport with a focus on the computational complexity of its resolution. Their main contribution is a result giving relation between accuracy (in term of error wrt the constraint) and number of iterations for Sinkhorn (and their) algorithm. It show that a given accuracy can be obtained with a complexity near linear to the size of the problem which is impressive. Using the in-depth computational study the authors propose a new algorithm with the same theoretical property of Sinkhorn but more efficient in practice. very small numerical experiment suggest that the propose GreenKhorn method works in practice better than Sinkhorn. This paper is very interesting and propose a theoretical computational study that can be of huge help to the community. It also explain why the recent entropic regularization works so well in practice and when it shouldn't be used. The new Greedy Sinkhorn algorithm is very interesting and seem to work very well in practice. I still have a few comment that I would like to be addressed in the rebuttal before giving a definite rating. Comments: - The whole paper seem a bit of a mess with theorems given between definitions and proofs scattered everywhere except below the theorem. I understand the the authors wanted to tell a story but in practice it makes the paper more difficult to read. Beginning with theorem 1 is ok and finishing with its proof is also OK. But the remaining should be done in classical math stuff theorem 2 followed by proof then theorem 3 that uses theorem 2 and so on. - One of the take home message from theorem 1 is actually not that good: if you want to decrease the l1 error on the constraint by 10, you will need to perform 10^3=1000 times more iterations. this also explain why sinkhorn is known to converge rather quickly to a good enough solution but might take forever to actually converge to numerical precision. It should have been discussed a bit more. Actually I would really like to see numerical experiments that illustrate the O(1/k^{1/3}) convergence of the error. - The propose Greenkhron is very interesting but it has strong similarity with the stochastic sinkhorn of [GCPB16]. My opinion is that the proposed method is better because it basically update only the variable that violate the most the constraints whereas stochastic will spend time updating already valid constraints since all variables are updated. Still good scientific experiments require the authors to provide these comparison and the fact that the code for [GCPB16] is available online gives the authors no excuse.

Reviewer 3



*** Post rebuttal : Sounds good to me! *** This paper shows that, with appropriate choice of parameters, the Cuturi-Sinkhorn algorithm, and a greedy variant called Greenkhorn, both calculate an approximate solution to the optimal transport problem in near linear-time in the input size. I like the clean analysis which is quite easy to follow, and with the power of hindsight, it is clear that the 1-norm (or TV) is the right error metric for entropic penalties (especially because of Pinsker, ie -entropy is strongly convex wrt TV norm). Minor question: In the statement of theorem 1, when the runtime is actually O(n^2 + S), which is O(max(n^2,S)), why do the authors prefer to report the runtime as O(n^2S) = O(n^2 L^3 log n/e^3)? They differ significantly when epsilon is a function of n, which might be desirable for statistical reasons (when the quantity actually being calculated is the empirical OT distance, but the quantity of interest is the population OT distance, we may not desire accuracy e to be a constant but instead decaying with n). Can the authors conjecture on whether the dependence on epsilon can be improved, or do the authors have some reason to believe that the dependence is essentially optimal? Please clarify in the notation section what r_i(F) and c_j(F) mean, they are not introduced before being used in Algorithm 2,3. I obviously can guess what they mean, but it's easier for everyone if this is added. Nevertheless, this algorithm and associated result are quite nice, and the proof of theorem 1 is so clean and intuitive. (however, the forward reference in the proof of theorem 3 to theorem 5,6 suggests that some reorganization may be beneficial) I think the authors may want to rename some of their "theorems" as "propositions" or "corollaries". The one or two central results in the paper (which even a non-expert should be tempted to read) should be labeled theorems, it is not good mathematical writing style to have 6 theorems in a short paper, all of which are essentially parts of the same goal -- it's like the use of bold or italics, if it is overused, it loses its effect of catching the reader's attention. Overall, I like the paper a lot, it seems correct, it is clean and not unnecessarily complicated, it is important, the empirical results are promising, and if the paper is reorganized and rewritten a bit, it can be a great nips paper.